# Hyaluronic Acid-Based Nanosystems for CD44 Mediated Anti-Inflammatory and Antinociceptive Activity

**DOI:** 10.3390/ijms24087286

**Published:** 2023-04-14

**Authors:** Saniya Salathia, Maria Rosa Gigliobianco, Cristina Casadidio, Piera Di Martino, Roberta Censi

**Affiliations:** 1School of Pharmacy, Università di Camerino, 62032 Camerino, Italy; 2Department of Pharmacy, Università “G. d’Annunzio” di Chieti e Pescara, 66100 Chieti, Italy

**Keywords:** hyaluronic acid, CD44, nanosystems, antinociceptive, anti-inflammatory

## Abstract

The nervous and immune systems go hand in hand in causing inflammation and pain. However, the two are not mutually exclusive. While some diseases cause inflammation, others are caused by it. Macrophages play an important role in modulating inflammation to trigger neuropathic pain. Hyaluronic acid (HA) is a naturally occurring glycosaminoglycan that has a well-known ability to bind with the cluster of differentiation 44 (CD44) receptor on classically activated M1 macrophages. Resolving inflammation by varying the molecular weight of HA is a debated concept. HA-based drug delivery nanosystems such as nanohydrogels and nanoemulsions, targeting macrophages can be used to relieve pain and inflammation by loading antinociceptive drugs and enhancing the effect of anti-inflammatory drugs. This review will discuss the ongoing research on HA-based drug delivery nanosystems regarding their antinociceptive and anti-inflammatory effects.

## 1. Introduction

In response to environmental factors and noxious stimuli, the body uses pain as a defence mechanism. Pain is a proactive beneficial immune response in the acute phase, but neuropathic pain becomes problematic in the chronic phase. The nociceptive sensory neurons (nociceptors) activate the neuropathic pain signal, but the immune system also plays a significant role in defining the active bidirectional crosstalk between pain and inflammation [1]. Nociceptors can control innate and adaptive immune functions by releasing neuropeptides and neurotransmitters [2,3,4,5]. In response, neuronal plasticity and chronic pain can be controlled by mediators (lipids, cytokines, and growth factors) released by the immune cells [6,7,8]. The signals and messages sent by the nervous system are propagated in milliseconds. This is theorised to be partly why nociceptors are ideally positioned to be first responders to pathogens and tissue injury. Nociceptors release neuropeptides in adverse situations that activate the macrophages of the immune system to control neuropathic pain and inflammation. Increasing evidence from studies shows that macrophages can induce and resolve pain via macrophage–nociceptor interactions [9,10,11].

Inflammation is a complicated process. It was previously known to be a response to infection; however, in recent years, inflammation was found to cause multiple diseases: atherosclerosis [12], depression [13], Alzheimer’s [14], obesity [15], etc. [16,17]. The elevation of inflammatory markers (C-reactive protein) or release of pro-inflammatory cytokines can be detected to confirm its presence [18]. Even a minimal increase in the expression of these markers increases the risk of inflammation, which is abnormal without an externally harmful stimulus. An unhealthy lifestyle can be a cause of abnormal inflammation [19,20,21].

However, inflammation, in some cases, can be due to easily defined causes such as gastritis [22], arthritis [23], neurodegenerative diseases [24] and sepsis [25], where inflammation is called a necessary evil. This is likely a part of the first line of defence, but it is also necessary to keep it in check before chronic derelict harm is caused to the host [26].

Derived from monocytes (M0), macrophages can polarise, becoming either pro-inflammatory or classically activated (M1), or anti-inflammatory or alternatively activated (M2) [27]. M1 macrophages activate pro-inflammatory cytokines and chemokines, which initiate and modulate the inflammatory immune response [28].

M1 macrophages have a flat, round cell shape, whereas M2 macrophages are elongated. The macrophage phenotype varies depending on the environmental stimulus as McWhorter et al. proved in a study in which they tested if the elongation of cells could manipulate the macrophage phenotype [27]. M0 exposed to elongated channels expressed arginase-1 (Arg-1), a marker for M2, and those exposed to wider channels expressed inducible nitric oxide synthase (iNOS), a marker for M1. The modulation of cytokines due to factors such as viruses [29], infections [30] and fibrosis [31] can also manipulate macrophage phenotype. Extensive studies are being conducted to imitate body functions and cytokine expressions to obtain a better understanding of the complicated mechanism of action of macrophage polarisation.

Recent technology and brainstorming have led to several nanosystem designs that can directly target M1 for its antinociceptive activities. Ligand-specific nanosystems can directly target individual macrophage receptors to manipulate cytokine release, modulate phenotype expression and protect the cargo from clearance by the reticuloendothelial system (RES) [32,33].

Cluster of differentiation 44 (CD44) is a glycoprotein receptor that is heavily expressed on the surface of macrophages and tumour cells. It is also a well-known receptor for hyaluronic acid (HA) [34,35], which is a large natural polysaccharide. Hence, the use of HA has been widely studied in cancer research. The chemical composition of HA is repeating units of D-glucuronic acid and N–acetyl-d-glucosamine [33]. A hydrogen bond is formed between the C6-hydroxy group of HA with an N-terminus of CD44 to stabilize the binding [36].

CD44 receptor is present in all immune cells but its binding with HA is dependent on the homeostatic conditions of the body [37]. HA is heavily present in the extracellular matrix (ECM) during homeostatic conditions due to its ability to retain water [38,39]. Under these conditions, alveolar macrophages (displaying a distinct hybrid M1/M2 phenotype [40]) are the only immune cells to bind to HA [37]. However, under inflammatory conditions, reactive oxygen species and nitrogen species break down HA into smaller fragments [41] that undergo phagocytosis by macrophages via CD44-mediated uptake. Lee-Sayer et al., theorised that the binding of HA with CD44 during inflammation assists in keeping the macrophages at the site of inflammation and further aids in their function [37]. Therefore, there is minimal HA-CD44 binding under homeostatic conditions.

M1 macrophages have the highest surface presence of the CD44 receptor of all phenotypes [42]. Since chronically inflamed tissue shows a consistent presence of M1, studies were performed to design drug delivery systems that target the CD44 receptors of these M1 to polarise their phenotype to M2 [43].

‘Nanoparticles’ is an umbrella term for nanohydrogels, self-assembling nanosystems, nanoemulsions, nanocomposites, etc. Nanoparticles of HA can be altered in many ways, including structurally and chemically, for drug-loading purposes, surface modifications, transdermal delivery or nanoparticle uptake at the target site. Particle size affects the mode of cellular uptake and the efficiency with which they can pass through the body without being cleared by the RES, lungs, liver or spleen. If the particle size is small (10–20 nm), particles are less likely to be taken up by macrophages, which is a problem when targeting macrophages, for example, during therapy for pain and inflammation. The clearance rate for large particles (>1 μm) is also high, since they tend to aggregate. Therefore, it is suggested that the particle size for drug delivery should be between 20 nm and 1 μm [44,45,46].

In this review, we focus on the exploitation of HA-based drug delivery nanosystems for optimal CD44 targeting to suppress acute and chronic inflammation and, subsequently, neuropathic pain.

## 2. HA and Inflammation: Influence of Molecular Weight

Different molecular weights (MW) of HA are present in all biological tissues and fluids [47] (Figure 1). Indigenously, high-molecular-weight (HMW) HA is found, which is then degraded into smaller fragments of low molecular weight (LMW) depending on the environmental factors [48,49]. This degradation of HA is essential for a number of bodily functions, for instance, as a lubricant in the synovial fluid [48,50].

The manipulation of MW of HA in drug delivery systems can lead to analgesic, anti-inflammatory and immunostimulatory results [48,49,50,51]. Although there is a lack of detailed research on the specificity of the antinociceptive effects of HA of different MW, it is widely accepted that HMW HA inhibits the activation of lipopolysaccharide (LPS) by directly binding to the toll-like receptor-4 (TLR-4) under inflammatory conditions [52].

To maintain proper functioning, there needs to be a balance between the quantity of HA being produced and degraded in the body [48]. The MW of HA is controlled through the body by shifting between its cellular uptake and degradation under homeostatic conditions. The enzymes hyaluronidases control the degradation of HA [53,54]. Two types of hyaluronidases (HYAL), HYAL1 and HYAL2, are involved in the active degradation of HA. While HYAL1 targets LMW HA, HYAL2 is known for breaking down HMW HA chains to 20 kDa [53].

There is a long-running debate in the research regarding the pro/anti-inflammatory properties of HAs of different molecular weights. Studies, including that of Isa et al., have suggested that HMW HA is shown to have anti-inflammatory properties by inhibiting the production of interleukin-1β, one of the more prominent inflammatory cytokines, and that LMW HA is a promoter of inflammation [55,56,57,58,59,60]. Baeva et al. provided the explanation that the breakdown of longer chains of HMW HA by HYAL2 produces HAs with fewer disaccharides of LMW HA that accumulate at inflamed sites to activate the nuclear factor kappa-light-chain-enhancer of the activated B cells (NF-κB) pathway [58] (Figure 2). However, results from the HA hydrogel osteoarthritis (OA) therapy study by Agas et al. showed that LMW HA (37,900 Da) promotes anti-inflammatory properties. The LMW HA hydrogel significantly lowered the expressions of pro-inflammatory cytokines, tumour necrosis factor-alpha (TNF-α) and interleukin (IL)-1 [59]. Chernos et al. performed anti-inflammatory studies on human cell lines with butyrylated derivatives of LMW HA to provide optimal visco-supplements for OA therapy [56]. Chistyakov et al. also noted that long-term exposure to LMW HA can suppress the inflammation induced by the TNF-α pathway [61]. Inflammation is a complex mechanism that involves two major human systems, the nervous system and the immune system. Ongoing research is attempting to unravel the mysteries that surround the molecular weights of HA and their effect on pain and inflammation; however, there is no sure way to say which molecular weight causes, and which resolves, inflammation at present.

## 3. HA-Based Nanosystems

HA-based nanosystems can be categorised in a number of ways, such as nanohydrogels, nanoparticles and self-assembling nanosystems (Figure 3). A variety of delivery system designs have been considered for this review that display the vast potential of using HA to counter inflammation, which is notoriously at the root of many chronic diseases.

### 3.1. Drug Delivery Systems

Drug delivery systems describe how the drugs are carried into and throughout the body. The following studies use HA to load drugs that are further tested in vitro and/or in vivo.

Nanohydrogels are three-dimensional (3D)-polymeric networks of nanoscale dimensions, with a crosslinked structure that gives them potential flexibility and versatile behaviour [62,63]. They have the dual advantage of hydrogels, allowing for the high encapsulation efficiency of hydrophilic compounds, and of nanostructures, allowing for high cellular internalisation [64]. Environmental stimuli such as temperature and pH can be used to develop site-specific nanoparticles which makes them an optimal choice for novel theranostic applications [65,66,67].

Quagliariello et al. synthesized a quercetin-loaded HA nanogel for its anti-inflammatory effect in breast tumour cells [68]. The 200 kDa HA that was used provided protection to the drug from oxidative and enzymatic degradation in the tumour environment. A solvent–non-solvent method was used for synthesis, with glutaraldehyde as a crosslinker. The drug-loaded nanogel showed a size of 211 nm. Free HA was noted as having insignificant cytotoxicity; however, the crosslinker in the nanogel showed a 10–20% cytotoxic effect. The expressions of anti-inflammatory cytokines (IL-8, IL-6 and IL-19) decreased by up to 55% with nanogel when compared to the control. A 30–40% increase in anti-oxidative effect was observed when quercetin was co-loaded with everolimus. The group concluded that HA-nanohydrogels provide an excellent template for studying tumour microenvironments, opening perspectives for further studies.

Barbarisi et al. tested the effect of co-loading quercetin and temozolomide in HA-nanogel as a therapy with an anti-inflammatory effect in glioblastoma tumour cells [69]. The solvent–non solvent method was used to make nanogel with 200 kDa HA and a glutaraldehyde crosslinker. The drug-loaded nanogel had a size of 197 nm and a ζ-potential of −31.3 mV. Just like in their previous study [68], the group noted cytotoxic effects from the crosslinker in the nanogel in this research. HA-induced, receptor-mediated endocytosis was noted with 30% nanogel internalization after 2 h. HA on the surface was used for this nanosystem to avoid opsonization by the RES and provide a longer drug retention time.

While both of the above-mentioned studies [68,69] successfully synthesized anti-inflammation-promoting nanogels, it should be noted that the glutaraldehyde used to stabilize these nanogels was responsible for increasing their cytotoxicity. Further studies need to be performed, which can either suggest a less toxic crosslinker or a formulation method that does not require a crosslinker for stabilization.

Storozhylova et al. were looking for an efficient drug delivery system that showed a longer retention rate for the treatment of inflammatory joint diseases [70]. The group synthesised in situ forming, non-crosslinked HA-fibrin hydrogels, containing HA-nanocapsules co-loaded with dexamethasone and galectin-3 inhibitor. The drug-loaded nanocapsules showed the suppression of inflammation after intra-articular administration. However, the study noted that further investigation was required to treat chronic synovial inflammation.

It is beneficial to develop a drug delivery system with positive results in terms of efficiency, but it is even better to design a system with a hassle-free administration route. Even though injectable nanohydrogels are considered non-invasive techniques, transdermal drug deliveries take this definition one step further. As previously mentioned, transdermal drug delivery (TDD) has classically more successfully been associated with the use of nanoparticles. TDD is a painless method of delivering therapeutics onto intact skin [71,72]. The nanosize [73], drug retention [74], and drug release rates [75] of polymeric nanohydrogels are winning factors in their wide use as a targeted TDD method, and nanohydrogel size is essential for successful skin penetration. Nanohydrogels can be manipulated into loading drug-loaded nanocapsules that successfully penetrate the skin and intake water. This leads to the swelling of the nanocapsules and subsequent drug release [76,77,78].

Wei et al. tested the anti-inflammatory effect of topically administered HA nanohydrogels with baicalin–nanocrystals (NC) [79]. An 800−100 kDa HA nanogel was used to assist the skin permeability of poorly soluble baicalin. Four *w*/*v* concentrations of HA were used to optimise the nanogel: 0.5%, 1%, 1.5% and 2%. The increase in HA concentration led to an increase in the viscosity and elasticity of the nanogel; however, it also saw a decrease in the drug release rate and skin permeation rate. The 1% *w*/*v* HA was chosen as the optimal concentration, with a 20-fold increase in skin permeability as compared to the control. The size of 1% *w*/*v* HA nanogel was 193 nm.

The biodegradability of HA allows for homogenous drug distribution in the gel matrix [80,81]. Liu et al. used electrospinning to make an absorbable nanofibrous hydrogel for wound healing under chronic diabetic conditions [82] (Figure 4). A 1400 kDa thioether grafted HA, crosslinked with Fe^3+^ (FHHA-S/Fe) nanogel, was synthesized for the purpose of wound healing by modulating the site of injury. Overall, the crosslinking increased the stability of the nanofibres by two-fold. Complete absorption of the nanogel was observed at 72 h. The thioether grafting increased IL-4 expression, leading to 33% and 18% faster wound healing as compared to the non-ether nanogel. A 24% decrease in the expression of M1 macrophages was observed, along with a 22% increase in the expression of M2 macrophages after treatment with HMW HA nanogel.

Just like previously mentioned studies [68,69], this research [82] also used a crosslinker to stabilize their nanogel. However, Fe^3+^ showed no cytotoxic effects. It also provided antibacterial properties to the nanosystem.

Pleguezuelos-Villa et al. synthesized mangiferin-loaded HA-based nanoemulsions for their anti-inflammatory effect on skin lesions [83]. Two ranges of the MW of HA were tested for the nanoemulsions, 40–50 kDa (LMW) and 1000–1200 kDa (HMW). The LMW HA nanoemulsion size was detected at 221 nm, and for HMW it was 393 nm. A gradual, sustained release of the drug was found in all nanosystems after 24 h but LMW HA nanoemulsion with surfactant was the highest, with 10–15%. The different MW of HA did not affect the oedema inhibition for the anti-inflammatory activity of the nanoemulsion. However, the use of a surfactant decreased the oedema inhibition activity of the nanoemulsion. HMW HA only seemed to affect the size of the nanoemulsion.

Manca et al. used curcumin-loaded HA vesicles (hyalurosomes) for skin lesions [84]. This study evolved the field of vesicles by synthesizing hyalurosomes. The group enhanced the properties of conventional liquid vesicles by including a gel-core structure to provide more stability and a nanosized diameter to make them nanovesicles. The organic solvent-free polymer dispersion method was used for the synthesis. Two *w*/*v* concentrations of 200–400 kDa HA were tested: 0.1% and 0.5%. The concentration of 0.1% showed a size of 166 nm and an encapsulation efficiency of 76%, whereas that of 0.5% showed a size of 157 nm and a curcumin encapsulation efficiency of 79%. The enhanced nanovesicle structure increased the encapsulation by 10–13% as compared to conventional liposomal vesicles. However, 0.5% *w*/*v* hyalurosomes showed a higher viscosity than 0.1% and more stiffness, by 14%, with the addition of curcumin. An increased improvement in induced skin lesions in in vivo tests was seen in 0.5% *w*/*v* hyalurosomes. Re-epithelialized skin was also observed by day 6.

Yang et al. synthesized HA nanostructured lipid carriers (NLCs) loaded with ropivacaine (RVC) and dexmedetomidine (DMDT) [85]. NLCs are evolved lipid nanoparticles that include both liquid and solid lipids that decrease the order of crystal arrangement of conventional lipid nanoparticles to provide a higher drug loading efficiency [86]. The group used the solvent diffusion method to create an HA-based (3 kDa) drug delivery system that increased the duration of the analgesic effects of RVC and DMDT by 75%. The NLCs had a size of 108 nm, a ζ-potential of −30.7 mV due to the presence of HA, and drug encapsulation efficiency of 89.5% for RVC and 88.1% for DMDT. The HA NLCs also increased the cell viability of the drugs from 61.2% (free drug solution) to 80%. The group also performed in vivo skin permeation tests that showed that the encapsulation of the drugs in the NLCs increased their permeability by 67%. The NLCs also increased the antinociceptive effect of the drugs by 80 min. It was found that co-loading RVC and DMDT had a higher analgesic effect than loading a single drug.

Yue et al. also decided to formulate NLCs for TDD of their drug bupivacaine (BPV) for local anaesthesia [87]. The group used 300 kDa HA modified with linoleic acid and polyethylene glycol (PEG) for stealth properties. The NLCs were made with lipid melt-emulsification and solvent injection techniques that had a size of 154 nm, a ζ-potential of −40.1 mV and a BPV encapsulation efficiency of 88.9%. The NLCs showed a cell viability of 70% as compared to the 40% obtained by free drug solution. In vivo tests showed an increased antinociceptive effect by the NLCs as compared to free drug solution, by 50%.

Both the above-mentioned studies [85,87] used NLCs for their formulations; however, there were some notable differences between the two drug delivery systems. After 75 h of administration, one of the NLCs [85] showed a 70% antinociceptive activity, whereas this was only at 60% for the other NLC [87]. While this may not be a major difference, it is possible that the synergistic effect of co-loading two drugs favoured the former. The higher NLC size used by the latter can be attributed to their use of HMW HA.

Iannitti et al. tested the efficacy of an HA- and chondroitin sulfate (CS)-based medical device called Esoxx^®^ for the treatment of inflammation of the gastric mucosa, also known as gastritis [88]. The presence of HA and CS was seen to reduce inflammation and the discomfort that comes along with it in the tested patients. HA provides hydrophilicity to the submucosal connective tissue, which provides it with a better chance of healing. The group, however, concluded that further studies are needed, with a higher number of patients, for definitive results.

### 3.2. Macrophage Targeting Nanosystems

It has been well established that macrophages have an important role to play in modulating inflammatory responses and pain [10,11,43]. The pro-inflammatory M1-macrophage phenotype is responsible for the first line of defence, which is inflammation. Hence, for antinociceptive therapies, studies have developed nanosystems that target the CD44 receptor, which is heavily present on the surface of M1 macrophages. These therapies either work by lowering/inhibiting the effect of pro-inflammatory cytokines or polarising the macrophage phenotype from M1 to M2.

Zhang et al. used a layer-by-layer (LBL) NLC system for lidocaine (LA)-loaded chitosan and HA drug delivery systems [89] (Figure 5). LBL involves the alternative deposition of oppositely charged polyelectrolytes via electrostatic interaction for the assembly of multilayer films. This method decreases the drug release rate and enhances skin permeability. The group compared the characteristics of LBL-NLCs with simple NLCs. It is important to note that the size of the nanoparticles is a major deciding factor in whether the drug delivery system will reach the target site. The NLCs showed a size of 181 nm and a ζ-potential of +37.6 mV due to the outermost layer of chitosan. The in vivo tests for anaesthesia showed that NLCs had a 30% effect after 24 h of administration, whereas LBL-NLCs had an 80% effect. The combined biocompatibility of hyaluronic acid and chitosan provides a great template for the loading of drugs and target studies [90].

Farajzadeh et al. synthesized curcumin-loaded HA-polylactide (PLA) nanoparticles to test CD44-targeted antinociceptive activity and macrophage repolarization [43]. An HA of 20 kDa MW was used to prepare HA-PLA conjugates of 102 nm in size, with a ζ-potential of -24.5 mV and a curcumin encapsulation efficiency of 88%. Mouse peritoneal macrophages were used for in vitro studies that detected a burst release in the first 8 h of administration and a consistent subsequent release with 33% drug release after 144 h (6 days). The initial burst release could be due to the presence of the drug, which was closer to the surface of the nanoparticles. In vivo nanoparticle uptake was promoted by endocytosis; hence, it was necessary to stimulate the nanoparticles under similar conditions. Endocytosis is supported by an acidic environment, so the group also tested the drug release from the nanoparticles at a pH of 4.4. This test showed a 50% drug release after 120 h of administration. Markers for M1 and M2 macrophages (iNOS and Arg-1, respectively) were quantified to check macrophage repolarization. Curcumin-loaded nanoparticles increased Arg-1 expression by 50% as compared to free drug administration. This concluded the successful repolarization from M1 macrophages to M2 macrophages. The expressions of pro-inflammatory cytokines, TNF-α, IL-1β and IL-6, were also reduced by 86%, 85% and 87%, respectively, when compared to free drug administration.

Tran et al. also experimented with shifting the macrophage polarity as a means of therapy for inflammation [91]. However, instead of curcumin, they encapsulated plasmid deoxyribonucleic acid (pDNA) in their HA nanoparticles by modifying HA with a positively charged polymer poly(ethyleneimine) (PEI) (Figure 6). The pDNA expressed interleukin-4 (IL-4) and interleukin-10 (IL-10) genes that inhibit the production of TNF-α, which is a direct promoter of M1-expressed inflammatory cytokines. This reduction in the expression of TNF-α also leads to lower M1 polarisation and subsequently higher M2 polarisation [92]. However, the direct administration of IL-4 and IL-10 is reported to have toxic effects. Therefore, the group combined gene therapy with nanotechnology to synthesise a nanosystem that not only represses inflammatory cytokines but also polarises macrophages towards alternative activation.

Kosovrasti et al. targeted another M1-specific cytokine to reduce inflammation, TNF-α [93]. The group encapsulated TNF-α-specific, small, interfering RNA (siTNF-α) in HA nanoparticles. The formulation of the nanoparticles involved the blending of HA-PEI, HA-hexyl fatty acid and HA-PEG. The 78–90 nm diameter HA nanoparticles encapsulating siTNF-α reduced the level of production of LPS-induced TNF-α in macrophages, and hence reduced inflammation. The group concluded that this study could be beneficial when researching therapies for acute inflammatory diseases.

It is important to note that the nanosystems developed in two previously mentioned studies [43,91] were not tested in vivo, and both groups concluded that further studies were needed to ascertain the effect of their respective HA-based nanosystems under complex in vivo homeostatic conditions. However, the HA nanoparticles encapsulating siTNF-α by the group [93] were tested in vivo, and the expected anti-inflammatory results were found. However, they also noted that the results from one study should not be considered conclusive evidence.

Xie et al. used HA-containing ethosomes (ES) for the delivery of rhodamine B for TDD [94]. Ethosomes are a kind of liposomal vesicles that are known for increasing skin permeability, drug accumulation and targeting drug delivery [95,96,97] to avoid systemic toxicity. A 150 kDa HA was used to enhance drug entrapment in the ES using amphiphilic modifications to the HA backbone. Different ratios of HA:ES were tested for optimization: 2.5:1, 5:1 and 10:1. The increase in the ratio of HA led to an increase in the size of the drug delivery system, with a range of 593–916 nm according to dynamic light scattering (DLS); however, transmission electron microscopy (TEM) detected the size of the system to be <100 nm. This difference could be because TEM analysis is conducted on a dry sample and DLS is performed in water, which leads to swelling of the particles. A high HA concentration also led to an increase in the quantity of the encapsulated drug. In vivo tests were conducted, which noted skin permeation within 30 min of administration. HA was also noted to increase the effectiveness of the system by 30% compared to the drug delivery system without HA. Therefore, the 5:1 ratio was concluded to be optimal.

The specific targeting ability of HA and its transdermal absorption is paramount in its use of TDD [98]. HA is a major synovial fluid component and artificially administered HA provides temporary relief for OA [99]. Zerrillo et al. decided to take advantage of the low pH conditions in the synovial fluid and synthesize HA-loaded, pH-responsive poly(lactic-co-glycolic acid) (PLGA) nanoparticles with a triggered burst release for OA [100]. The therapeutic approach of HA on OA partly includes reducing the inflammation at the OA site. The burst release of drug in this study was triggered by the ammonium bicarbonate loaded in the PLGA nanoparticles. The 750–1000 kDa HA was injected into the PLGA nanoparticles that showed a size of 202 nm and 28% encapsulation efficiency. In vitro studies showed that PLGA-HA nanoparticles had a faster uptake than only PLGA nanoparticles. In vivo studies were also conducted, which showed that pH-responsive nanoparticles had a faster cargo release than non-pH-responsive nanoparticles. Fluorescence showed the presence of the nanoparticles in the knee even after 35 days of administration. The group concluded that a combined therapy of pH-responsive and non-pH-responsive nanoparticles would have a synergistic effect for optimal therapeutic conditions. Therefore, pH-responsive nanoparticles would have a burst release and non-pH-responsive nanoparticles would have a gradual, steady release of the drug.

Zerrillo et al. conducted another study where they tested HA-grafted PLGA nanoparticles for OA therapy [101]. Conventional HA therapy has a rapid clearance and short retention time. Grafting is used as a method to overcome these issues. In this case, a 20 kDa HA is used to synthesize a PLGA-HA copolymer and prepare PLGA-HA nanoparticles. The nanoparticles showed a size of 200 nm. Near-infrared dye tests showed that PLGA-HA nanoparticles had a 20% lower release rate after 10 days. There was a two-fold increase in the in vitro binding studies for PLGA-HA nanoparticles. Intra-articular injection was used for in vivo tests. After 48 h, PLGA-HA nanoparticles were noted to have penetrated the cartilage, unlike PLGA nanoparticles without HA.

Histochemical immunostaining has reported OA synovium to have a higher number of CD44 receptors than normal. This leads to an increase in inflammation due to the active presence of pro-inflammatory cytokines. This has further led to the increased targeting of CD44 as a therapy for OA using HA, a well-known CD44 ligand [102,103,104,105].

### 3.3. Self-Assembling Nanosystems

The molecular arrangement of disorganised components into ordered structures as a response to external stimulus is known as self-assembly. It is a phenomenon often found in nature. Biological nanostructures come self-assembled to form a DNA double-helix, cell membranes, peptide chains, etc. [106,107,108].

Vafaei et al. tested a budesonide (BDS)-loaded self-assembled HA nanosystem as a therapeutic agent for inflamed intestinal mucosa caused by inflammatory bowel disease (IBD) [109]. The self-assembling effect was enhanced by amphiphilic chemical modifications to the HA backbone. The 10 and 25 kDa HAs were tested and the thin-film hydration method was used to load BDS. The human colon carcinoma cell line was used for in vitro tests. The increased degree of chemical modification led to a decrease in the size of nanoparticles. The size for 10 kDa HA decreased by 97 nm and, for 25 kDa HA, the size decreased by 61 nm.

Mota et al. synthesized a PLGA-loaded HA hybrid systems for viscosupplementation in OA [110]. The group used 1500–1800 kDa HA, 45–75 kDa PLGA and a modified spontaneous emulsification/solvent diffusion method for synthesis. Oleic acid was also used to propagate long-term controlled drug release and provide stability to the nanosystem. DLS was used to check the size of HA-loaded PLGA particles (373 nm) and oleic-acid-modified particles (4561 nm). The increase in the size of particles with oleic acid was attributed to particle agglomeration. Since DLS cannot differentiate between particle agglomeration and a large size, atomic force microscopy (AFM) was used to check the size of oleic-acid-modified particles, which showed a particle size of 409 nm. The use of the hybrid system for HA administration increased the drug release rate to up to 8 h, as compared to the instant dissolution of free HA. The in vivo anti-inflammatory effect was tested for free HA solution (76.9%) and HA-PLGA particles (82.6%).

Currently, the most common commercially available treatment method of OA is the intra-articular injection of HMW HA. The therapy shows results but is short-lived owing to the fact that HMW HA is prone to active HYAL-mediated degradation [48,53,111]. Kang et al. realised the potential of using self-assembling HA nanoparticles as an alternative therapy for OA [111] (Figure 7). The group chemically modified the backbone of 10 kDa HA with cholanic acid to create amphiphilic HA nanoparticles. The 221 nm nanoparticles were tested in vivo and in vitro, where they showed a remarkable improvement compared to conventional HMW HA OA therapy. In vitro studies noted that HA-nanoparticles had a larger CD44-mediated uptake and cartilage penetration, of up to 41 μm, than free HA. Intra-articular injections were used for in vivo studies, where it was observed that HA-nanoparticles blocked the CD44 receptor and prevented further cartilage degeneration in OA-induced mice, unlike free HMW HA. The attenuated NF-κB pathway activity was also noted to prevent pro-inflammatory cytokine expression. The group concluded that empty amphiphilic HA-nanoparticles showed an increased resistance to HYAL degradation compared to free HA. This study has the potential to be continued with loading HA-nanoparticles to increase efficacy.

El-Refaie et al. also tested the use of hyalusomes; however, they made them self-assembling TDD hyalusomes for use in OA [112]. The film hydration technique was used to make 1% *w*/*v* HA (8–11.7 kDa) hyalusomes. The elasticity of the system was increased by the addition of ethanol. Ethanol-modified gel-core hyalusomes showed a size of 226 nm with an encapsulation efficiency of 32.6%. Ex vivo results showed that the skin permeability of gel-core hyalusomes was increased by 5.5-fold as compared to 1% HA solution. In vivo results also favoured a six-fold increase in HA in joint tissues after gel-core hyalusome administration.

## 4. Hyaluronic Acid Nanosystems, Study Gaps and Future Directions

The active involvement of HA in inflammation makes it a top contender for use as a nanosized drug carrier. However, its negative charge is critical for in vivo use. Several studies have reported that positively charged particles are more prone to causing the secretion of cytokines, inducing more T-cell proliferation, which leads to the recruitment of cytotoxic T cells, directly damaging the erythrocyte membrane, and eliciting an immune response [113,114,115,116,117,118].

The rapid evolution of nanotechnology has changed the fundamental chemical, physical and physiological aspects of a successful drug delivery system [119]. Nanosystems come in different shapes and sizes, such as nanohydrogels, nanoparticles, nanocomposites, and self-assembling nanoparticles, among others. With their high drug encapsulation efficiency and biological efficacy, they have become the top contenders for research on therapeutics, diagnostics, and imaging. Table 1 summarises the HA-based nanosystems referred to in this review.

In this review, we highlight the recent advancements in HA-based nanosystems being developed for antinociceptive and anti-inflammatory activities. Providing an overview of the mechanism of action of pain and inflammation, we consolidated some HA-based nanosystems that are being used as drugs and encapsulating agents to target CD44 receptor on macrophages for therapy for various disorders, such as gastritis [90], IBD [109], breast cancer [68], glioblastoma [69], inflammatory joint disease [70,112], wound healing [82], local anaesthetic [87], OA [100,101,110,111,112] and skin lesions [83,84]. These drug-encapsulating nanosystems have shown increased pharmacokinetic properties as compared to solo drug administration due to their biological versatility and targeting ability.

The size of the nanosystem is important for successful targeted drug delivery action. Nanosystems of <100 nm are rapidly cleared by the RES and macromolecules are cleared by the kidney and the spleen. The HA-based nanosystems mentioned above fall within the preferred range of drug delivery systems that can favourably avoid clearance by the body. The hydrophilicity of HA provides the nanosystem with a stealth effect that avoids opsonization [120].

Along with the advantages of using HA, it is also important not to overlook the associated challenges. The effect of the molecular weight of HA on inflammation is not completely understood. Broadly, it is accepted that HMW HA holds anti-inflammatory properties and LMW HA is pro-inflammatory [55,56]. However, Zerrillo et al. found anti-inflammatory results with LMW HA [100,101]. When designing and synthesising nanosystems, LMW HA is preferred as it is easy to work with. LMW HA also produces smaller size nanosystems that are more effectively hidden from clearance by the immune system. As Kang et al. reported, HMW HA is degraded by HYAL, which can be avoided with LMW HA [111]. Further studies are required to clearly lay out the role of different MWs of HA in the functioning of the human body.

The delivery method of the nanosystem to the target area also affects the drug design. Therapeutics is moving toward simpler, non-invasive drug delivery designs. This, however, can pose a limitation when synthesising the nanosystem. For instance, more and more research is being conducted on the advancement of TDD systems. The anionic nature of HA poses a problem in this case. The hydrophilicity of HA reduces its rate of skin permeability [121,122]. Although this can be overcome either by hydrophobic modifications to the HA chain or by coating it with a hydrophobic compound (such as chitosan), it still poses a challenge, as it may not be a feasible step for the proposed function of the nanosystem.

Most of the nanosystems mentioned in this review are in the concept stage. Some have only undergone in vitro experimentation, while others conclude that further in vivo studies are required. No conclusive result can be provided until more research is conducted. There is a gap in the understanding of the working of pro- and anti-inflammatory cytokines that control macrophage polarisation. Obtaining a clearer picture of the biological mechanisms that are involved will help to optimise the nanosystems for more targeted action. Furthermore, the preclinical drug delivery models need to be tested for biodegradability regarding their reduced toxicity, absorbability and high retention rate for a prolonged effect.

Translating biomaterials from laboratory experiments to commercial development takes decades [123,124]. Newer inventions must undergo a longer safety protocol to become approved. A medical device that is “substantially equivalent” to one already on the market will find it easier to obtain Food and Drug Administration (FDA) approval than a newer one [125,126,127]. Therefore, while lab experiments continue to provide positive results, further rigorous testing will need to be undertaken for years before it becomes commercially available.

## 5. Conclusions

HA presents multiple beneficial characteristics, such as biocompatibility, hydrophilicity, biodegradability, non-toxicity, and binding ability with the CD44 receptor, which make it an optimal choice for usage in anti-inflammatory drug delivery systems. Studies have shown that it can be formulated as a hydrogel, nanocomposite, or nanoparticle, or superficially and chemically modified for wider targetability. Combining it with other compounds (PLGA, chitosan, etc.), enhances its chemical (amphiphilicity) and biological (resistance to HYAL degradation) properties, improves the efficacy of the nanosystems, and increases its in vivo half-life, leading to prolonged drug release [128,129]. While there further tests and designs are needed in the proposed nanosystems, the results have been fairly positive, with HA displaying high anti-inflammatory properties and CD44 targeting abilities.

## Figures and Tables

**Figure 1 ijms-24-07286-f001:**
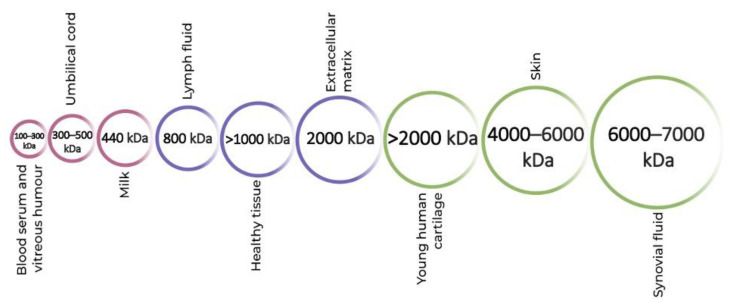
Molecular weights of hyaluronic acid in different parts of the human body.

**Figure 2 ijms-24-07286-f002:**
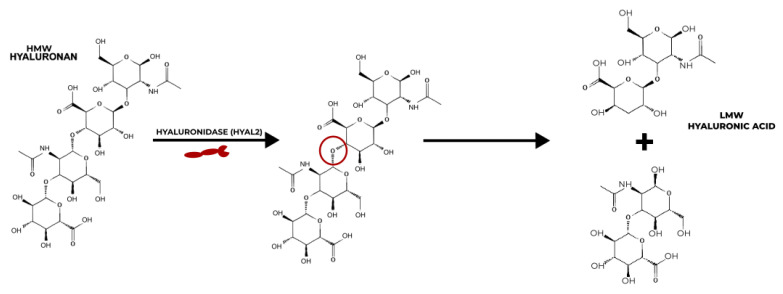
Mechanism of action of HYAL2 on the degradation of HMW HA to LMW HA.

**Figure 3 ijms-24-07286-f003:**
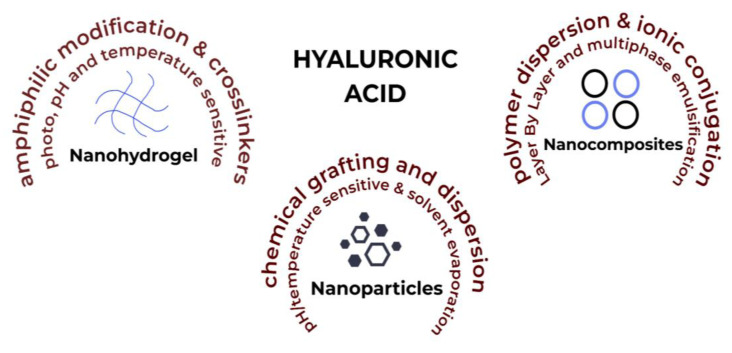
HA-based nanosystems with factors affecting their synthesis and production.

**Figure 4 ijms-24-07286-f004:**
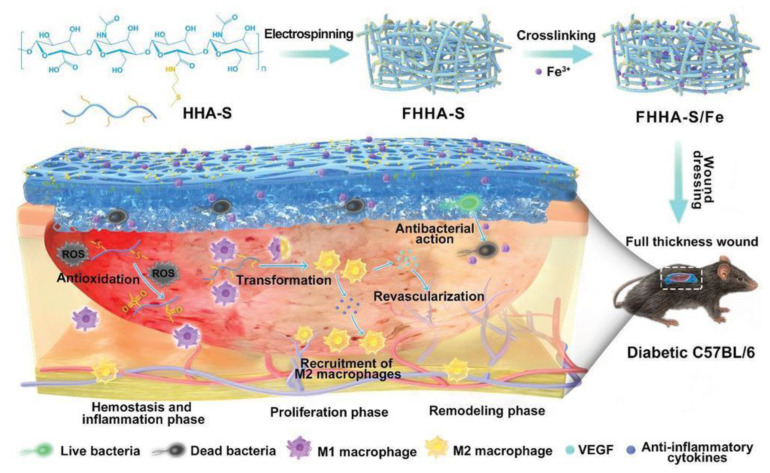
Schematic illustration of the absorbable thioether grafted hyaluronic acid nanofibrous hydrogel for synergistic modulation of the inflammation microenvironment to accelerate chronic diabetic wound healing. Illustration of the preparation procedure of FHHA-S/Fe, dressing of FHHA-S/Fe on full-thickness wound model in diabetic C57BL/6 mouse, and the mechanism of FHHA-S/Fe for enhanced chronic wound healing effect. Copyright Wiley-VCH Verlag GmbH. Reprinted with permission from [82].

**Figure 5 ijms-24-07286-f005:**
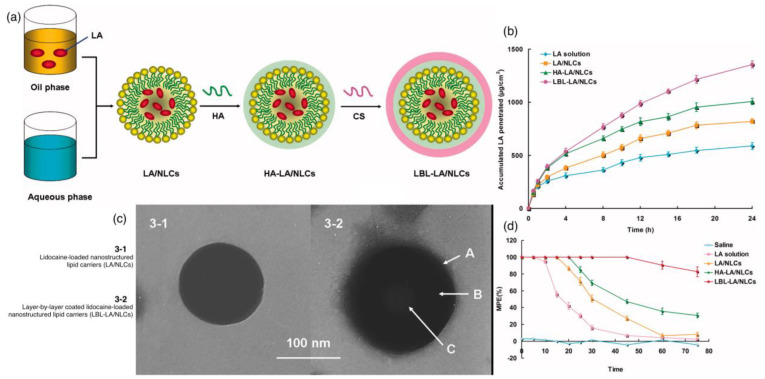
(**a**) Scheme of the fabrication of LBL-LA/NLCs. (**b**) In vitro permeation profiles of LA from different formulations. (**c**) TEM images of the structural morphology of the LBL-LA/NLCs and LA/NLCs. Image 3-2 shows different shades of gray that indicate the multiple layers of coating from C being the innermost and A being the outermost. (**d**) In vivo TFL test for the evaluation of the local anesthetic effects of LA-containing formulations. Adapted with permission from [89].

**Figure 6 ijms-24-07286-f006:**
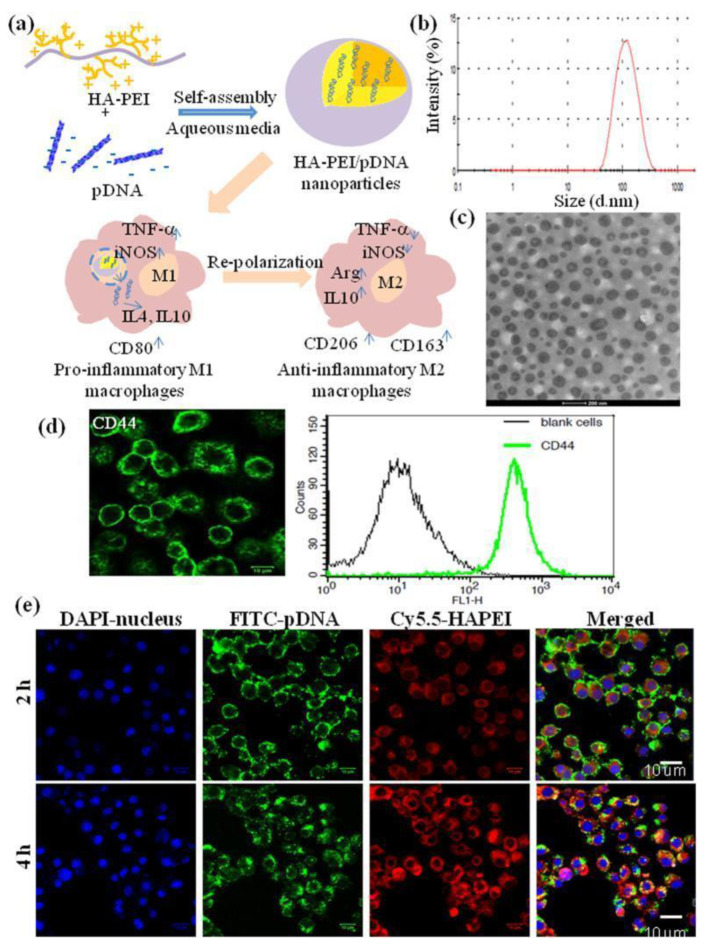
(**a**) Schematic illustration of pDNA encapsulation into HA-PEI nanoparticles for the re-polarization of pro-inflammatory M1 macrophages to anti-inflammatory M2 macrophages. (**b**) Size distribution of HA-PEI/pDNA (9:1) in PBS by DLS. (**c**) TEM image of HA-PEI/pDNA in PBS (9:1). (**d**) Confocal microscopy and FACS analysis of CD44 expression in J774A.1 macrophages. (**e**) Uptake of HA-PEI/pDNA nanoparticles in J774A.1 macrophages. Reprinted with permission from [91].

**Figure 7 ijms-24-07286-f007:**
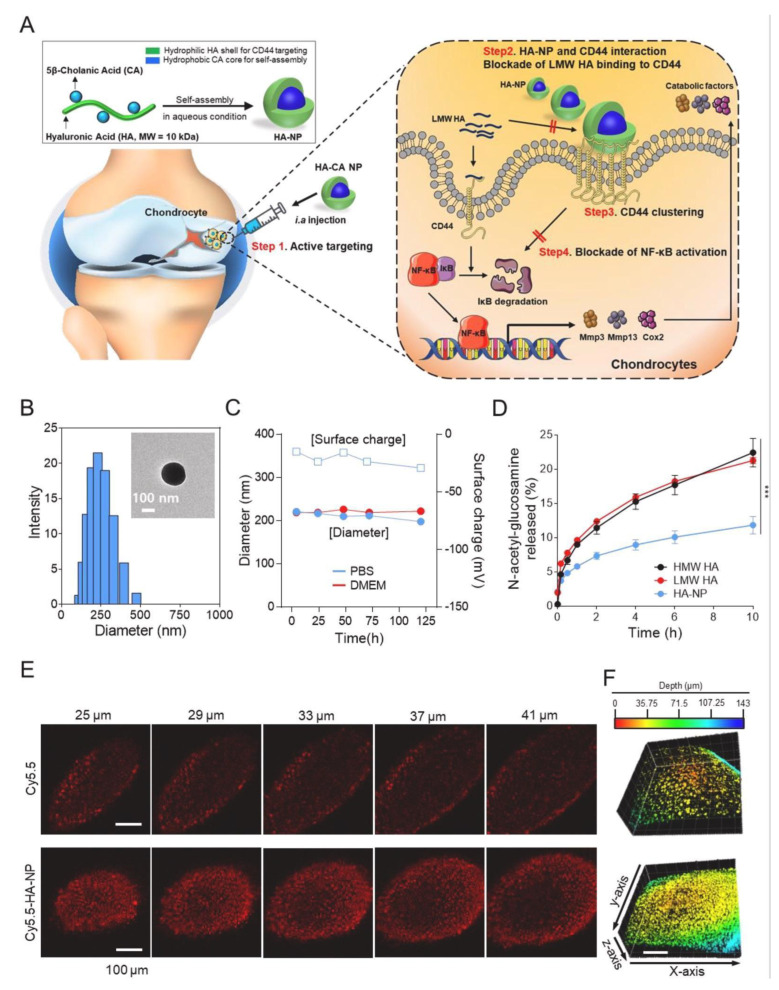
Characteristics of HA-NPs. (**A**) Schematic illustration of HA-NPs for treatment of OA. (**B**) TEM images and size distribution of HA-NP. Scale bar, 100 nm. (**C**) Time-dependent changes in particle size and surface charge of HA-NP in PBS and DMEM. Data are presented as mean ± SEM (n = 5). (**D**) Generation of N-acetyl-glucosamine after treatment of 1 mg/mL free HAs (10 kDa LMW and 2000 kDa HMW) or HA-NP with 100 IU/ml HYAL-II. Data are presented as mean ± SEM (n = 4). *** *p* < 0.001. (**E**) Representative serial images (25–41 μm depth at intervals of 4 μm) from the femoral cartilages after i.a. injection of Cy5.5 and Cy5.5-labeled HA-NP into normal mice. Scale bars, 100 μm. (**F**) Three-dimensional lateral view of the femoral cartilages after i.a. injection of Cy5.5 and Cy5.5-labeled HA-NP into normal mice. Scale bars, 100 μm. Reprinted with permission from [111].

**Table 1 ijms-24-07286-t001:** Summary of HA-based nanosystems discussed in the review.

Nanosystem	HA in Nanosystem	Administration Route	Formulation Method	Characterisation	Drug	Therapy	Reference
Drug Delivery Systems							
HA-nanogel loading quercetin	200 kDa Sodium hyaluronate	-	Solvent–non solvent method	Size: 211.3 nm ζ: −35.8 ± 1.3 mV	quercetin	Anti-inflammatory effect in breast cancer tumour cells	[68]
HA-nanogel loading quercetin	200 kDa Sodium hyaluronate	-	Solvent–non solvent method	Size: 197.3 ± 3.3 nm ζ: −31.3 ± 1.1 mV	quercetin	Anti-inflammatory effect in glioblastoma tumour cells	[69]
Non-crosslinked HA-fibrin hydrogels containing HA-shell nanocapsules co-loaded with dexamethasone and galectin-3 inhibitor	40 kDa, 700 kDa, 1.5 MDa Sodium hyaluronate	Intra-articular	Solvent displacement method	Size: 122–135 nm ζ: −29 ± 5 mV	dexamethasone and galectin-3 inhibitor	Therapy for inflammatory joint diseases	[70]
HA nanocrystal hydrogels loading baicalin	800–1000 kDa Sodium hyaluronate	TDD system	Homogenization	Size: 189.21 ± 0.36 nm	baicalin	anti-inflammation	[79]
Thioether-grafted HA nanofibrous hydrogel	1400 kDa Sodium hyaluronate, thioether modified and crosslinked with Fe^3+^	TDD system	Electrospinning	Size: 60 ± 11 nm	HMW HA	wound healing in diabetic conditions	[82]
HA-based mangiferin nanoemulsion	40-50 kDa and 1–1.2 MDa HA	TDD system	Emulsion method	Size: 296 nm ζ: −30 mV	mangiferin	Anti-inflammation for skin lesions	[83]
Curcumin-loaded HA-nanovesicles	200–400 kDa Sodium Hyaluronate	TDD system	Organic solvent-free dispersion method	Size: 157–166 nm ζ: 24 ± 4 mV	curcumin	Anti-inflammation for skin lesions	[84]
HA nanostructured lipid carriers loading ropivacaine and dexmedetomidine	3 kDa HA, PEG-DSPE modified	TDD system	Solvent diffusion method	Size: 108.2 ± 3.3 nm ζ: −30.7 ± 2.8 mV	Ropivacaine and dexmedetomidine	Local analgesic	[85]
HA-modified nanostructured lipid carriers loading bupivacaine	300 kDa HA	Local administration	Emulsification and solvent injection methods	Size: 154.6 ± 5.1 nm ζ: −40.1 ± 3.9 mV	bupivacaine	Local anaesthetic	[87]
HA-chondroitin sulfate medical device (Esoxx)	-	Oral delivery	-	-	Esoxx^®^	Gastritis	[88]
Macrophage-Targeting Nanosystems							
HA-polylactide nanoparticles encapsulating curcumin	20 kDa Sodium hyaluronate	-	Solvent evaporation method	Size: 102.5 nm ζ: −24.5 ± 2.2 mV	Curcumin	Macrophage repolarisation	[43]
HA-chitosan lipid nanostructures loading lidocaine (layer-by-layer)	-	TDD system	Melt-emulsification method	Size: 181.2 nm ζ: +37.6 ± 4.2 mV	Lidocaine	Local anaesthesia	[89]
HA-nanoparticles loading pDNA	20 kDa Sodium hyaluronate	in vitro transfection	Coupling reaction	Size: 185.9 nm ζ: −11.6 mV	pDNA	Macrophage repolarisation	[91]
HA-nanoparticles encapsulating siTNFα	20 kDa Sodium hyaluronate	Transfection	-	Size: 85–110 nm	TNF-α specific small interfering RNA	Inhibiting LPS-induced inflammation	[93]
HA containing ethosomes encapsulating rhodamine	150 kDa Sodium hyaluronate	TDD system	Homogenization	Size: 593.8 nm ζ: +10 mV	Rhodamine	Quick, high-efficiency TDD system	[94]
pH-responsive HA-loaded PLGA nanoparticles	750–1000 kDa Sodium hyaluronate	Intra-articular injection	Single-emulsion solvent evaporation method	Size: 202.7 ± 2.3 nm ζ: −21.0 mV	HMW HA	Osteoarthritis	[100]
HA decorated on PLGA nanoparticle surface	21–40 kDa Sodium hyaluronate	Intra-articular injection	Double-emulsion solvent method	Size: 200 ± 2 nm ζ: −23 ± 2 mV	HMW HA	Osteoarthritis	[101]
Self-Assembling Nanosystems							
HA-nanoparticles loading BDS	10–25 kDa Sodium hyaluronate, DA modified for amphiphilicity	in vitro dynamic dialysis drug release study	Thin-film hydration method	Size: 207 ± 11 nm ζ: −14.56 ± 0.22 mV	Budesonide	IBD-induced pain/inflammation	[109]
HA-PLGA hybrid systems	1.5–1.8 MDa Sodium hyaluronate	Viscosupplementation	Spontaneous emulsification solvent diffusion method	Size: 373 ± 270 nm ζ: −16.65 mV	HMW HA	Osteoarthritis	[110]
Amphiphilic HA-nanoparticles	10 kDa HA	Intra-articular injection	Chemical conjugation	Size: 221 ± 1 nm ζ: 15.08 ± 0.83 mV	LMW HA	OA treatment	[111]
Gel-core HA nanovesicles	8–11.7 kDa HA	TDD system	Thin layer evaporation technique	Size: 232.8 ± 7.2 nm ζ: −45.1 ± 8.3 mV	HMW HA	OA therapy	[112]

## Data Availability

Not applicable.

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
