# Peer review of "Hyaluronic Acid-Based Nanosystems for CD44 Mediated Anti-Inflammatory and Antinociceptive Activity"

_ijms, 2023, doi:10.3390/ijms24087286_

Round 1

Reviewer 1 Report

Due to the advantages of high drug encapsulation efficiency, good biocompatibility and targeting, nanosystems have been widely used in drug development, therapeutics and other fields. The positive role of negatively charged HA in the inflammatory response makes it an excellent carrier for nanomedicines, especially the ability of HA to bind to CD44 on the surface of macrophages, which makes hyaluronic acid-based nanosystems capable of targeting CD44. Therefore, it is the best choice for anti-inflammatory drug delivery system. This review discusses the investigation of the antinociceptive and anti-inflammatory effects of HA-based drug delivery nanosystems. Effective cutting-edge information compiled for the development of HA-based nanosystems in the direction of immune inflammation.

The following minor queries should be handled by the authors:

1.In lines 33-35, the introduction to " nociceptors are positioned to be first responders to pathogens and tissue injury" is not very relevant to the described interaction between the immune system and nociceptors, it is recommended to delete or change a more in-depth introduction to the interrelationship between pain and inflammation.

2.In the second paragraph of the introduction, it is introduced that the occurrence of inflammation can lead to many diseases, but there are few examples. Examples should be added to highlight the harmfulness of inflammation. Here, it is recommended to cite these two documents to supplement the introduction of inflammation.

( Doi: 10.1016/j.mtbio.2022.100215;10.1002/advs.202204999)

3. In line 80, the chemical structure of HA is introduced in the text, so it is suggested to add what is the specific binding domain of HA and CD44, or what is the way they bind

4. The second part is titled "HA for CD44 Targeting: Influence of Molecular Weight". However, the second part introduces that HA with different molecular weights can produce different pro-inflammatory or anti-inflammatory effects. The effect of the molecular weight of HA on its targeting of CD44 was not clearly indicated.

5. In lines 393-405, when introducing the research of Zerrillo et al, the focus is on its absorption rate, drug release rate, etc., rather than macrophage targeting.

6. In the fourth part, paragraphs 1-3 introduce the advantages of HA and nanotechnology. Paragraphs 4 and 5 point out the problems in development. However, paragraph 6 is changed to introduce the advantages of HA-based nanosystems. The layout of paragraphs confusion.

7. Some punctuation marks are used incorrectly in the article, for example, there is an extra "." before [35, 36] in line 87, and before [52-57] in line 138.

8. The content in the introduction is too long and can be simplified.

9. The pictures in the text are not uniformly labeled. For example, the labels in Figure 5 are: a), b), c)... The labels in Figure 6 are: (a) (b) (c)... It is recommended to agree to the labeling.

Author Response

Response to the Editor and the Reviewers

Manuscript Number: ijms-2279964

Dear Editor,

Thank you for the opportunity to revise our manuscript entitled “Hyaluronic Acid-Based Nanosystems for CD44 Mediated Anti-Inflammatory and Antinociceptive Activity”.

Please find below the list of modifications we made to improve our manuscript according to the Reviewers´ comments.

Best Regards,

Piera Di Martino

Reviewer 1

Due to the advantages of high drug encapsulation efficiency, good biocompatibility and targeting, nanosystems have been widely used in drug development, therapeutics and other fields. The positive role of negatively charged HA in the inflammatory response makes it an excellent carrier for nanomedicines, especially the ability of HA to bind to CD44 on the surface of macrophages, which makes hyaluronic acid-based nanosystems capable of targeting CD44. Therefore, it is the best choice for anti-inflammatory drug delivery system. This review discusses the investigation of the antinociceptive and anti-inflammatory effects of HA-based drug delivery nanosystems. Effective cutting-edge information compiled for the development of HA-based nanosystems in the direction of immune inflammation.

The following minor queries should be handled by the authors:

1.In lines 33-35, the introduction to " nociceptors are positioned to be first responders to pathogens and tissue injury" is not very relevant to the described interaction between the immune system and nociceptors, it is recommended to delete or change a more in-depth introduction to the interrelationship between pain and inflammation.

The authors thank the Reviewer for her/his effort in improving the manuscript. The authors have edited the relevant paragraph to make the connection between neuropathic pain and inflammation easier to understand.

2.In the second paragraph of the introduction, it is introduced that the occurrence of inflammation can lead to many diseases, but there are few examples. Examples should be added to highlight the harmfulness of inflammation. Here, it is recommended to cite these two documents to supplement the introduction of inflammation.

(Doi: 10.1016/j.mtbio.2022.100215;10.1002/advs.202204999)

The authors thank the Reviewer for her/his detailed analysis of the manuscript. The authors have cited the papers recommended by the Reviewer.

  1. In line 80, the chemical structure of HA is introduced in the text, so it is suggested to add what is the specific binding domain of HA and CD44, or what is the way they bind

The authors thank Reviewer 1 for her/his contribution to the in-depth analysis of our manuscript. The authors have considered this comment and inserted additional relevant information in the manuscript.

  1. The second part is titled "HA for CD44 Targeting: Influence of Molecular Weight". However, the second part introduces that HA with different molecular weights can produce different pro-inflammatory or anti-inflammatory effects. The effect of the molecular weight of HA on its targeting of CD44 was not clearly indicated.

The authors thank the Reviewer 1 for her/his careful revision of our manuscript. The authors have accordingly renamed the relevant section to fit the content.

  1. In lines 393-405, when introducing the research of Zerrillo et al, the focus is on its absorption rate, drug release rate, etc., rather than macrophage targeting.

The authors thank the Reviewer 1 for her/his careful revision of our manuscript. The authors have included appropriate information in the manuscript to provide clarity on the subject.

  1. In the fourth part, paragraphs 1-3 introduce the advantages of HA and nanotechnology. Paragraphs 4 and 5 point out the problems in development. However, paragraph 6 is changed to introduce the advantages of HA-based nanosystems. The layout of paragraphs confusion.

The authors thank the Reviewer for her/his comment. The comment was taken into consideration and the relevant paragraphs were rearranged to increase readability.

  1. Some punctuation marks are used incorrectly in the article, for example, there is an extra "." before [35, 36] in line 87, and before [52-57] in line 138.

The authors are grateful to Reviewer 1 for her/his in-depth evaluation of our manuscript. The suggested comments have been accepted and the punctuation has been corrected in the manuscript.

  1. The content in the introduction is too long and can be simplified.

The authors thank the Reviewer for her/his effort. The comment has been considered and the ‘introduction’ of the manuscript has been simplified.

  1. The pictures in the text are not uniformly labeled. For example, the labels in Figure 5 are: a), b), c)... The labels in Figure 6 are: (a) (b) (c)... It is recommended to agree to the labeling.

The authors thank the Reviewer 1 for her/his careful revision of our manuscript which improved its quality and readability. The recommended changes have been implemented and a consistent system of labelling has been adapted for the images in the manuscript.

Reviewer 2 Report

This paper overviewed the hyaluronic acid-based nanosystem and its role in CD44 Mediated anti-inflammatory and antinociceptive activity. It is interesting. Some small revisions need to be addressed.

1. In line 79, Abbreviation “HA” appears for the first time in the text, add the full name. Please check the use of all abbreviations in the main text.

2. Adjust the table 1 to fit the page width.

3. Remove the abbreviations at the end of the main text.

4. The article quotes some original drawings of references. Please determine whether you have obtained the right to copy.

5. Some sentences are not native enough, please polish them.

Author Response

Response to the Editor and the Reviewers

Manuscript Number: ijms-2279964

Dear Editor,

Thank you for the opportunity to revise our manuscript entitled “Hyaluronic Acid-Based Nanosystems for CD44 Mediated Anti-Inflammatory and Antinociceptive Activity”.

Please find below the list of modifications we made to improve our manuscript according to the Reviewers´ comments.

Best Regards,

Piera Di Martino

Reviewer 2

This paper overviewed the hyaluronic acid-based nanosystem and its role in CD44 Mediated anti-inflammatory and antinociceptive activity. It is interesting. Some small revisions need to be addressed.

  1. In line 79, Abbreviation “HA” appears for the first time in the text, add the full name. Please check the use of all abbreviations in the main text.

The authors thank the Reviewer for her/his comment. The authors accept the comment and have edited the text to include the full name of “HA” along with the abbreviation.

  1. Adjust the table 1 to fit the page width.

The authors thank the Reviewer for her/his effort in helping to increase the readability of the manuscript. As per the comments, the authors have adjusted the size of Table 1 to fit the page width.

  1. Remove the abbreviations at the end of the main text.

The authors thank the Reviewer for her/his comment and accordingly the abbreviations at the end of the main text have been removed.

  1. The article quotes some original drawings of references. Please determine whether you have obtained the right to copy.

The authors thank the Reviewer for her/his detailed analysis of the manuscript. The authors have edited the captions of the figures to make it clearer that all original images have been reprinted or modified with permission from respective journals.

  1. Some sentences are not native enough, please polish them.

The authors thank the Reviewer 2 for her/his comments on helping us improve the coherence of the manuscript. The authors have accordingly edited minor punctuation and grammar in the manuscript to make it easier to read.
